# Tumour and host cell PD-L1 is required to mediate suppression of anti-tumour immunity in mice

Janet Lau[1], Jeanne Cheung[1], Armando Navarro[2], Steve Lianoglou[3], Benjamin Haley[4], Klara Totpal[2], Laura Sanders[1], Hartmut Koeppen[5], Patrick Caplazi[5], Jacqueline McBride[6], Henry Chiu[7], Rebecca Hong[2], Jane Grogan[1], Vincent Javinal[2], Robert Yauch[8], Bryan Irving[1,†], Marcia Belvin[1], Ira Mellman[1], Jeong M. Kim[1,*] & Maike Schmidt[1,*]

Expression of PD-L1, the ligand for T-cell inhibitory receptor PD-1, is one key immunosuppressive mechanism by which cancer avoids eradication by the immune system. Therapeutic use of blocking antibodies to PD-L1 or its receptor PD-1 has produced unparalleled, durable clinical responses, with highest likelihood of response seen in patients whose tumour or immune cells express PD-L1 before therapy. The significance of PD-L1 expression in each cell type has emerged as a central and controversial unknown in the clinical development of immunotherapeutics. Using genetic deletion in preclinical mouse models, here we show that PD-L1 from disparate cellular sources, including tumour cells, myeloid or other immune cells can similarly modulate the degree of cytotoxic T-cell function and activity in the tumour microenvironment. PD-L1 expression in both the host and tumour compartment contribute to immune suppression in a non-redundant fashion, suggesting that both sources could be predictive of sensitivity to therapeutic agents targeting the PD-L1/PD-1 axis.

[1] Department of Cancer Immunology, Genentech, Inc., South San Francisco, California 94080, USA. [2] Department of Translational Oncology, Genentech, Inc., South San Francisco, California 94080, USA. [3] Department of Bioinformatics and Computational Biology, Genentech, Inc., South San Francisco, California 94080, USA. [4] Department of Molecular Biology, Genentech, Inc., South San Francisco, California 94080, USA. [5] Department of Pathology, Genentech, Inc., South San Francisco, California 94080, USA. [6] Department of Biomarker Development, Genentech, Inc., South San Francisco, California 94080, USA. [7] Department of Biochemical and Cellular Pharmacology, Genentech, Inc., South San Francisco, California 94080, USA. [8] Department of Discovery Oncology, Genentech, Inc. 1 DNA Way, South San Francisco, California 94080, USA. * These authors jointly supervised this work. † Present address: CytomX Therapeutics, Inc. 151 Oyster Point Blvd, Suite 400, South San Francisco, California 94080, USA. Correspondence and requests for materials should be addressed to J.M.K. (email: kim.jeong@gene.com) or to M.S. (email: maike.schmidt.sf@gmail.com).

Cancer cells elicit multiple mechanisms of immunosuppression to avoid obliteration by the immune system. Expression of PD-L1, a ligand for the T cell inhibitory receptor PD-1, plays a key role in attenuating anti-tumour responses in both mice and human cancer patients[1]. PD-L1 is thought to be adaptively expressed by tumour cells in response to inflammatory cytokines (for example, interferon-γ (IFNγ)[2]), thereby directly inhibiting T-cell-mediated killing[3–5]. Therapeutic use of blocking antibodies to either PD-L1 or PD-1 has produced unparalleled, durable clinical responses in a wide variety of solid and hematologic cancers[6–10], presumably by relieving suppression of primed T cells within the tumour microenvironment. Consistent with this concept is the finding that patients whose tumours express PD-L1 prior to treatment have a greater likelihood of response[6,11], best illustrated by the examples of non-small-cell lung cancer and metastatic urothelial bladder cancer[7,8,12,13]. However, one unexpected feature is that PD-L1 expression by infiltrating myeloid and other immune cells is more prevalent and can be even more predictive of response than PD-L1 expression by tumour cells alone[8,12]. The reasons for this are unclear but these data challenge the prevailing view that adaptive expression of PD-L1 by tumour cells is the sole source of PD-1 checkpoint control. Moreover, the significance of PD-L1 expression in tumours has emerged as a central and controversial unknown in the clinical development of immunotherapeutics in general, possibly contributing to the recent failure of a major phase III clinical trial in non-small cell lung cancer. Resolving the functional contributions of immune versus tumour cell PD-L1 expression will be critical to the continued progress of cancer immunotherapy.

Here we directly evaluate the relative roles of PD-L1 expression by the tumour and by the host's immune cells in the suppression of anti-tumour immune responses. Using genetic chimeras, we find that both tumour and host play non-redundant roles in regulating the PD-1 pathway, suggesting a key role for infiltrating immune cells in both generating and negatively regulating anti-tumour immunity.

## Results

**PD-L1 expression in human tumours and mouse models.** PD-L1 immunohistochemistry (IHC) analysis of human lung and breast tumours has identified three distinct patterns of positive PD-L1 expression: malignancies with predominant epithelial tumour cell PD-L1 expression, those with infiltrating immune cell expression only, or tumours with PD-L1 on tumour and immune cells (Fig. 1a,b). Although all three patterns can be predictive of response to therapy with anti-PD-L1 antibodies, the functional significance of PD-L1 expression by tumour versus immune cells is unknown and represents a major limitation to our understanding of how the PD-1/PD-L1 axis regulates the anti-cancer T cell response. To explore the relative contribution of the tumour and host compartment on PD-1-mediated immune suppression, we turned to preclinical models, as they are amenable to precise genetic deletion experiments. CT26 and MC38 are two immunogenic[14,15] colon tumour models that demonstrate PD-L1 expression on tumour cells as well as tumour infiltrating immune cells in vivo (Fig. 1c), with increased tumour PD-L1 expression following IFNγ exposure (Supplementary Fig. 1). Concordant with prevalent PD-L1 expression, both models were responsive to PD-L1 blocking antibodies (Fig. 1d,e), validating them as good models to test our hypothesis in subsequent genetic ablation studies.

**Genetic deletion of PD-L1 in tumour or host cells.** We next characterized tumour infiltrating immune cells in

PD-L1-deficient hosts (Supplementary Fig. 2) and the effect of this deficiency on tumour growth. Consistent with reports from LCMV-infected mice[16], absence of PD-L1 during T-cell priming in the lymph node led to increased cytotoxic T-cell infiltration and higher levels of activation markers when PD-L1 expressing tumours were inoculated in PD-L1-deficient mice (Fig. 2a). This finding is supported by transcriptional analysis of MC38 tumours in PD-L1-deficient hosts, in which gene sets representing various aspects of increased T-cell activation dominate the list of most significantly enriched sets (Fig. 2c; CAMERA false discovery rate (FDR) < 0.05). This increase in T-cell infiltration and activation was sufficient to trigger spontaneous complete regressions in 3/10 mice inoculated with MC38 tumours (Fig. 2b). Thus, despite continued expression of PD-L1 by the tumour cells (see below), the absence of PD-L1 expression by the tumour infiltrating host cells enhanced anti-tumour immunity, albeit not as well as the administration of anti-PD-L1 antibody (Fig. 1d).

Next, to determine the relative contribution of PD-L1 expression by tumour cells, CRISPR/Cas9 was used to genetically delete PD-L1 in the MC38 and CT26 cell lines. Individual cell clones with confirmed loss of PD-L1 expression (Supplementary Fig. 3b–e) showed comparable proliferative capacity to wild-type cells in vitro (Supplementary Fig. 3f,g), and readily formed tumours when injected subcutaneously into immune-deficient hosts (Fig. 2d, Supplementary Fig. 4a). Inoculation of PD-L1-deficient tumour cells into immune competent hosts, however, led to higher T-cell infiltration and activation marker expression, as seen for PD-L1-expressing tumours grown in PD-L1-knock out mice (Fig. 2a,e). In addition, approximately half of the tumour-bearing animals exhibited spontaneous regression of their tumours (4/10 and 8/10 for MC38 clones, 5/10 and 8/10 for CT26 clones; Fig. 2f and Supplementary Fig. 4b) despite continued expression of PD-L1 by the host. Although T-cell activation was comparable in the tumour and host knockout genotypes, transcriptional analysis of PD-L1-deficient tumours revealed a distinct enrichment of gene sets representing stromal/ECM remodelling and epithelial-meschenymal transition (Fig. 2g). Altogether, these results suggest that PD-L1 expression by both the tumour and host play distinct, partial roles in regulating anti-tumour immunity.

Although the absence of PD-L1 expression from either tumour or host cells produced similar increases in CD4 and CD8 T cell infiltration, but discrete transcriptional profiles, it was of interest to determine if there were other differences in tumour immune regulation. Indeed, we observed significant changes in cytokine gene expression that were also almost entirely distinct (Fig. 2h; Q value < 0.05). Tumours grown in PD-L1-deficient hosts were characterized by cytokines directly supporting lymphocyte infiltration such as CXCL9/10. By contrast, PD-L1-deficient tumours exhibited a different array of T-cell chemoattractants (e.g., CX3CL1), and an increase in general inflammatory cytokines, especially those associated with neutrophil/granulocytic MDSC infiltration, for example, CXCL1/3/5. Concordant with these alterations, the neutrophil/granulocytic MDSC population made up a larger proportion of the total immune infiltrate in PD-L1-deficient tumours (Fig. 2i), and the strong inflammatory response could be causative for the significant increase of gene sets representing stromal remodelling (Fig. 2g; CAMERA FDR < 0.05). Nevertheless, depletion of CD8 T-cells from mice bearing PDL1-deficient tumours completely abrogated tumour regressions (Fig. 2j), indicating that despite qualitative changes to the immune and stromal environment, the increased T-effector phenotype remained the main driver behind tumour rejections and interference with the PD-L1/PD-1 pathway. Altogether, these data suggest that lack of PD-L1 in the tumour

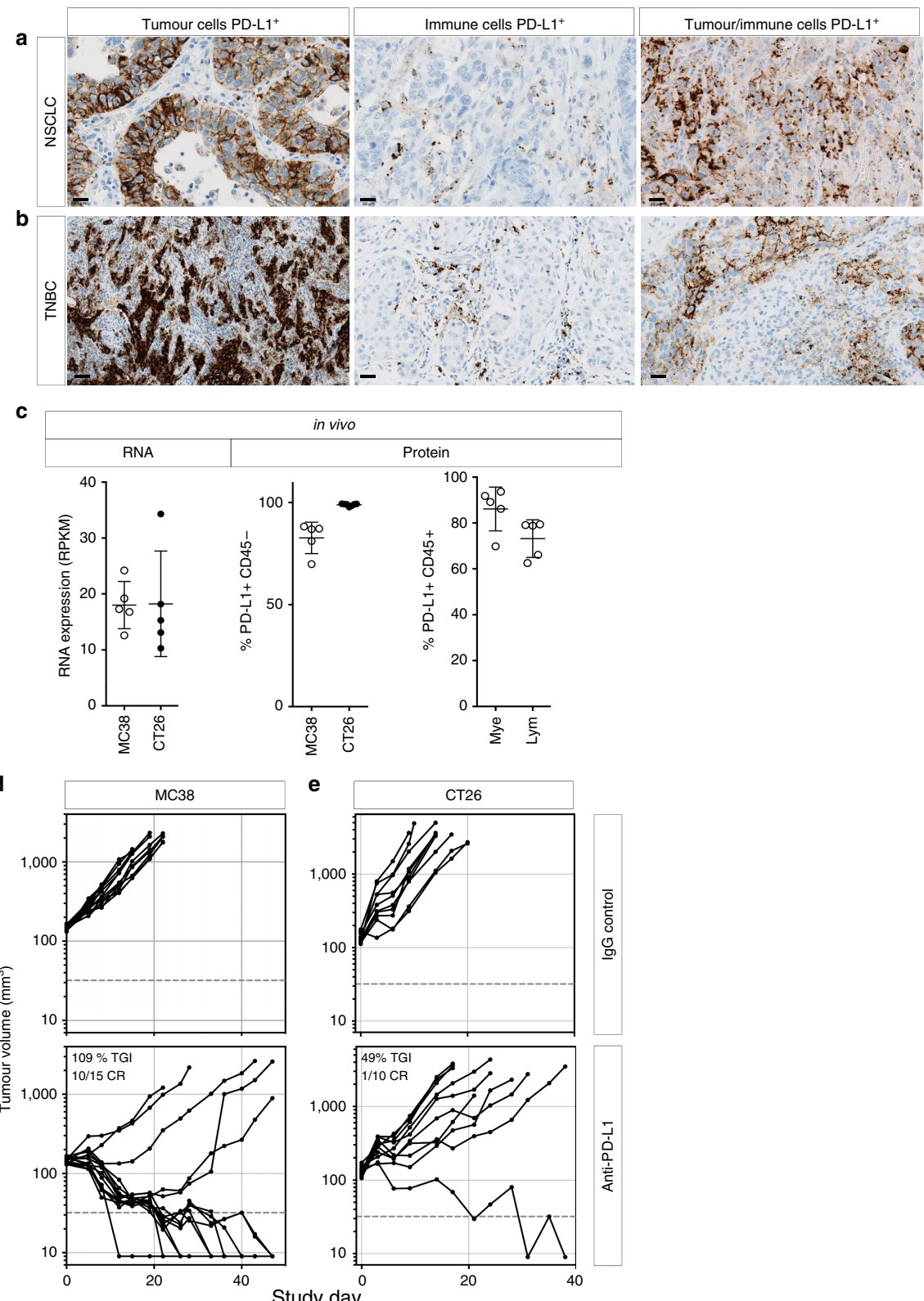

**Figure 1 | PD-L1 expression in malignant epithelial and immune cells of human tumours.** IHC analysis of human non-small-cell lung cancer (NSCLC) (**a**) and triple-negative breast cancer (TNBC) (**b**) samples identified three distinct patterns of PD-L1 expression (brown) in the tumour epithelium, immune cells or both compartments. In mouse tumour models *in vivo*, PD-L1 RNA and surface protein expression were detectable in MC-38 or CT-26 tumour cells, as well as in myeloid (mye) and lymphoid (lym) cells (**c**). Treatment of wild-type MC-38 (**d**) or CT-26 (**e**) tumours with anti-PD-L1 blocking antibodies resulted in slowed tumour growth and tumour regressions. If not labelled in graph, data shown is from MC38 (open circles) and CT26 (filled circles). Treatment data is representative of multiple independent study repeats with the same antibody. CR, complete regression; TGI, tumour growth inhibition. Scale bar represents 20 μm. Error bars depict s.d. from the mean.

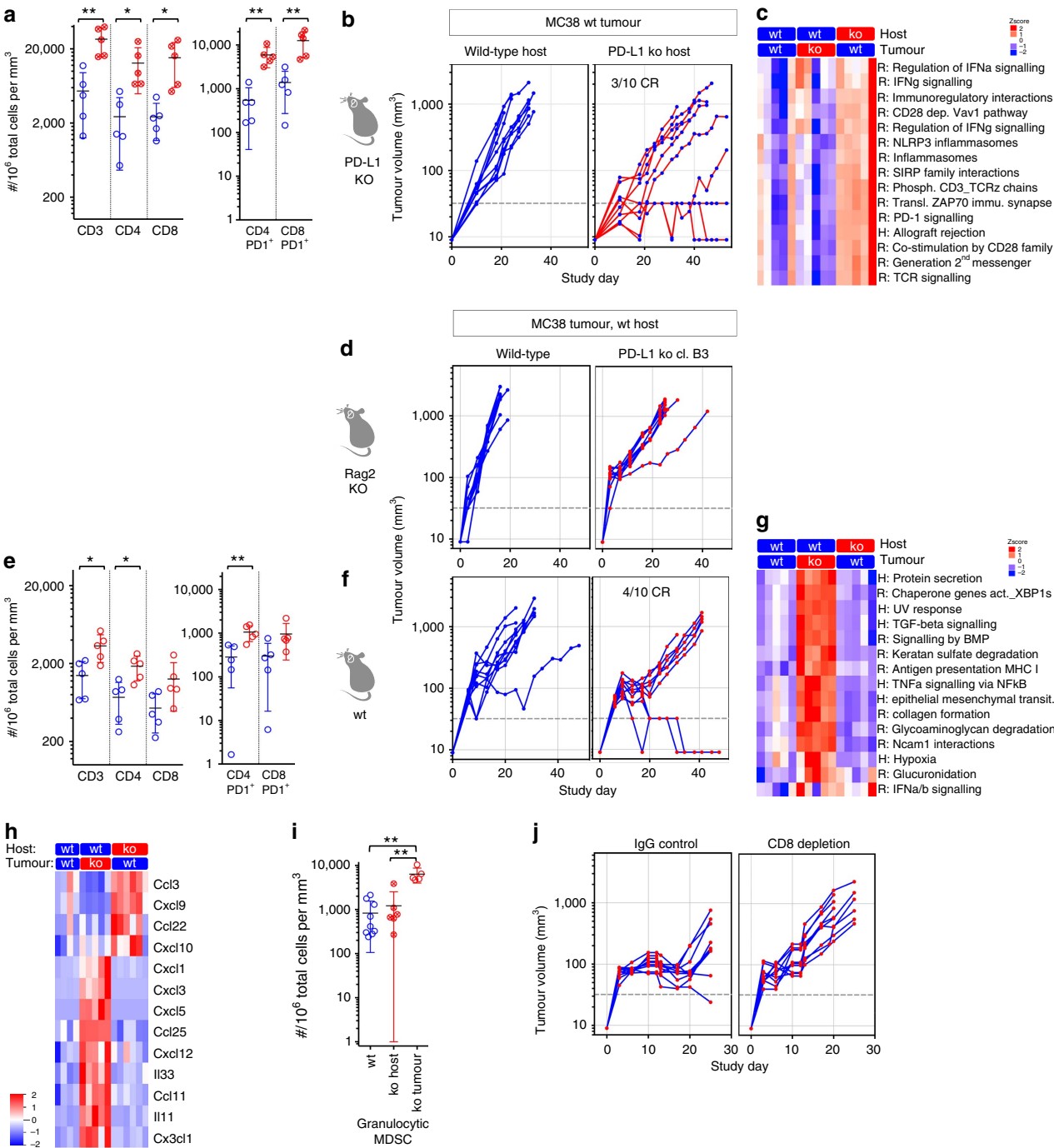

**Figure 2 | Genetic deletion of PD-L1 results in T-cell activation and tumor regression.** Flow cytometry analysis of PD-L1-deficient hosts showed increased T-cell infiltration and PD-1 expression in MC-38 tumours (**a**) and reduced tumour growth rate and complete tumour regressions (CR) were observed in 3/10 PD-L1-deficient mice (**b**). Most significantly upregulated gene sets were indicative of strong T-cell activation in MC38 wt tumor grown in PD-L1-deficient hosts (**c**). Similarly, wild-type (wt) mice inoculated with PD-L1-deficient MC-38 tumour cells showed increased activated, T-cell infiltration by flow cytometry (**e**). While PD-L1-deficient MC38 tumour cell inoculated into immune-deficient Rag2 KO mice showed normal tumour outgrowth (**d**), inoculation into wt mice leads to complete regression (CR) in 4/10 mice (**f**). Most significantly up-regulated gene sets were distinct and included stromal remodelling mechanisms (**g**). Significantly modulated chemokine gene expression was distinct between the host versus tumour PD-L1-deficient setting (**h**). Concordantly with the chemokine changes, increased numbers of granulocytic myeloid derived suppressor cells (MDSC) were detected in PD-L1-deficient tumours by flow cytometry (**i**). Depleting cytolytic CD8 + populations alleviated PD-L1-deficient tumour regression (**j**). Characterization of immune subsets by flow cytometry in tumours was performed between day 14–21 post inoculation. Tumour samples for RNA analysis were collected at d9 post inocculation. RNA and flow cytometry data shown is from wt MC-38 tumours in PD-L1-deficient host (crossed circles) or PD-L1-deficient MC-38 tumours in wt host (open circles), with PD-L1 wt status represented in blue and PD-L1 deficiency (ko) represented in red. For tumour growth curves, the symbol colour is representative of tumour cell PD-L1 status, the line colour for host PD-L1 status. Data are representative of a minimum of two independent study repeats. Statistical significance was determined by Student's *t*-test (\**P* < 0.05; \*\**P* < 0.01.) for individual RNA or protein analytes, by CAMERA FDR *P* < 0.05 for gene set enrichment analysis. Error bars depict s.d. from the mean. Mouse cartoon adapted from ref. 15.

or immune cells augments anti-tumour T-cell responses resulting in tumour clearance.

**PD-L1-mediated suppression by tumour cells is cell intrinsic.** To confirm that the observed phenotype was attributable directly to loss of PD-L1 in tumour cells, we introduced an inducible

mouse PD-L1 complementary DNA (cDNA) into PD-L1-deficient tumour cell clones. Ectopic expression of PD-L1 in deficient clones was sufficient to prevent tumour rejections and re-established *in vivo* growth similar to the wild-type parent line (Fig. 3a,b). To further understand if cell intrinsic PD-L1 was required for direct cell-to-cell inhibition or was able to establish a general immune suppressive tumour microenvironment, we

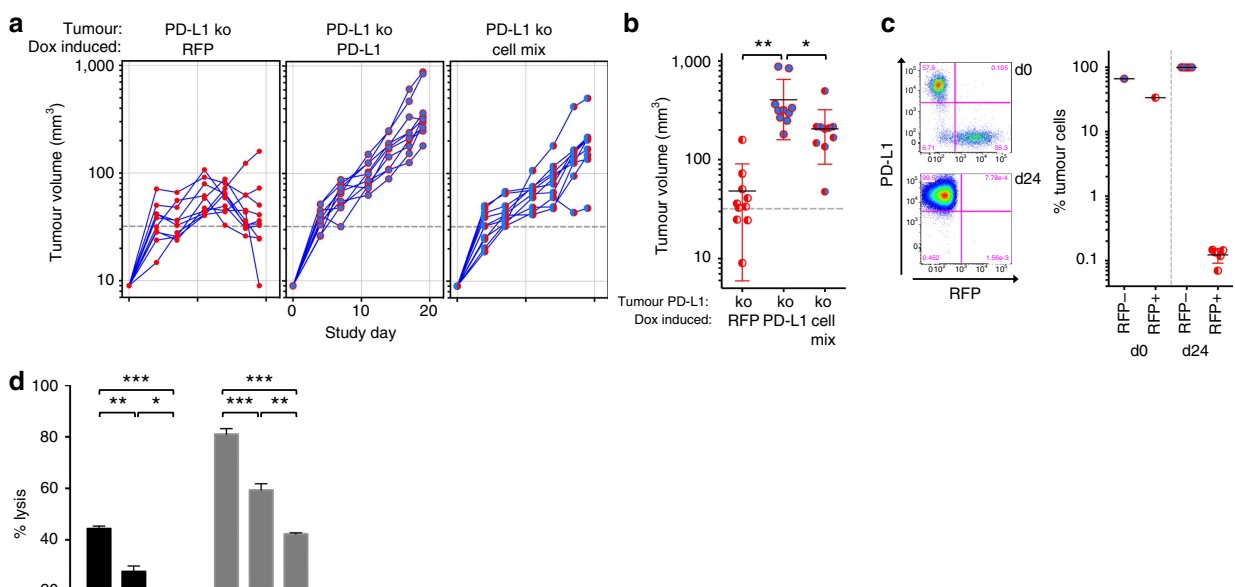

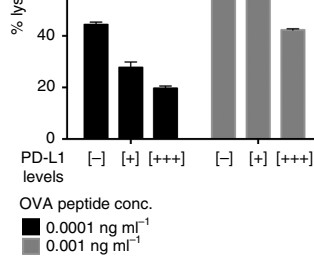

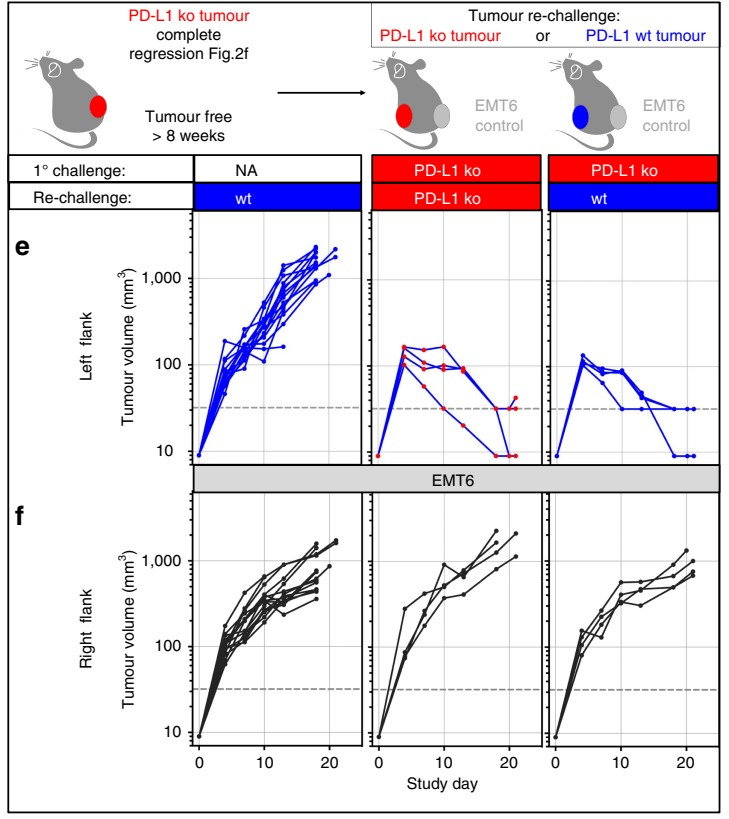

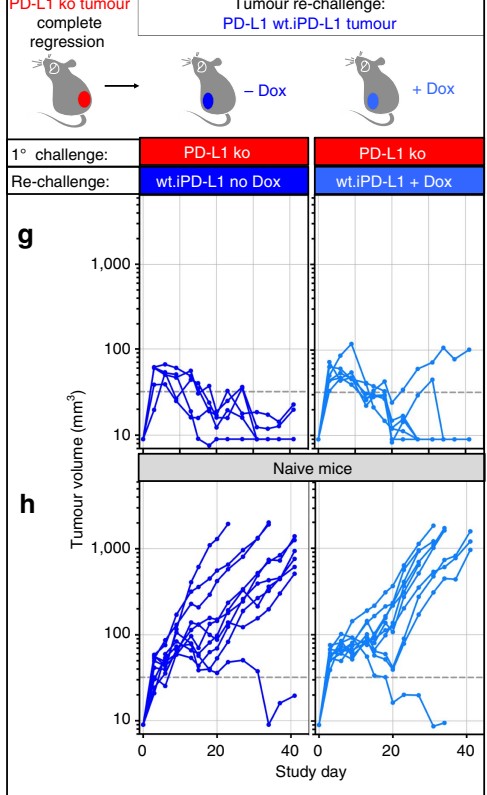

inoculated mice with a mix of PD-L1-deficient and PD-L1-enabled tumour cells. After an initial phase of stasis, mixed tumours established *in vivo* growth with similar kinetics to PD-L1-enabled tumours (Fig. 3a right panel, b). While both PD-L1-deficient and PD-L1-enabled tumour cells were detectable at the day of inoculation, tumours collected at study endpoint were solely comprised of PD-L1-enabled tumour cells (Fig. 3c). These data suggest that PD-L1 is directly required at the tumour–T-cell interface, where the antigen stimulus is delivered, to suppress tumour cell killing, and does not appear to be sufficient to create an immune suppressive microenvironment that could protect other tumour cells in trans.

We further investigated the dose dependency and potency of this direct cell protective mechanism in an *in vitro* setting. We exposed antigen-specific cytolytic T cells to cognate antigen-positive target cells expressing varying levels of PD-L1 *in vitro*. While lack of PD-L1 on target cells led to a high degree of target cell lysis, PD-L1 was able to partially suppress T-cell-mediated cell lysis in a dose dependent manner (Fig. 3d). These results suggest that PD-L1 expression on target cells directly protects tumour cells from cytolytic T cell killing.

**PD-L1 in the context of antigen-experienced T cells**. The PD-1/PD-L1 pathway plays a key role in negatively regulating T-cell activity, characterized by reduced effector cell function and failure to acquire T-cell memory[17]. As ablation of PD-L1 from tumour cells acted to reinvigorate T-cell cytotoxicity, we next addressed if memory formation was also promoted in the absence of tumour cell-derived PD-L1. Mice with complete regression of PD-L1-deficient CT26 tumours were re-challenged with the same PD-L1-deficient cell line following a two-month tumour free period. Compared to age-matched naive, or mice inoculated with tumour cells for the first time, CD4+ and CD8+ T-cell proliferation was significantly higher in animals experiencing re-challenge, and changes in CD8+ effector memory cell phenotype were consistent with a T-cell memory response to antigen re-exposure (Supplementary Fig. 5c,e; Student's *t*-test $P < 0.05$). Similarly, mice re-challenged after initial complete regression showed rapid tumour rejection when re-challenged with the same PD-L1-deficient tumour cells (Fig. 3e), while control tumours on the contralateral side progressively grew, comparable to growth in naive mice (Fig. 3f). Notably, productive clearance of PD-L1-deficient tumours induced a memory response against wild-type, PD-L1-expressing tumours on secondary challenge (Fig. 3e right panel). These data suggest that endogenous PD-L1 levels were not sufficient to protect the transplanted tumours from the re-called T-cell response. Similar results were obtained with MC38 tumour cells that over-expressed PD-L1 following dox-induction *in vivo* (Fig. 3g),

despite super-physiological PD-L1 levels (Supplementary Fig. 4d). These observations suggest that the ability of PD-L1 to enable tumour establishment *in vivo* depends on the balance between timing and magnitude of the developing anti-tumour immune response.

**Analysis of additional immune modulatory mechanisms**. Next we performed experiments to understand why a subset of PD-L1-deficient tumours escape and sustain growth despite increased immune activation. To investigate the immune evasion mechanisms employed by PD-L1-deficient tumours, we analysed outgrowing PD-L1-deficient tumours by RNA profiling, flow cytometry and IHC analysis. Despite their exponential growth, PD-L1-deficient tumours that escaped immune surveillance showed higher levels of apoptotic gene expression (Supplementary Fig. 6a), reduced tumour cell viability and reduced proliferation (Supplementary Fig. 6b,d) when compared to size-matched wild-type tumours. Thus, outgrowing tumours appeared to remain under partial control by the immune system, but adapted their immune suppressive mechanisms to enable tumour growth.

Several mechanisms employed by tumour cells to evade immune surveillance have been described, such as the down-regulation of MHC class I on the cell surface to reduce antigen presentation[18–20]. Indeed, PD-L1-deficient tumour cells displayed reduced MHC class I levels on their cell surface, and a lower percentage of cells were positive for MHC class I expression in the non-immune compartment (Fig. 4a). MHC class I expression by the tumour cell lines *in vitro* was unchanged, indicating that the lower surface levels *in vivo* was due to adaptive mechanisms to the tumour microenvironment (Supplementary Fig. 7).

As a substantial subset of tumour cells remained MHC I positive, we next investigated if additional immune suppressive mechanisms could contribute to tumour escape in PD-L1-deficient tumours. We observed increases in tumour cells expressing the immune regulatory B7 family member PD-L2, the second inhibitory ligand that mediates T-cell suppression through PD-1 (Fig. 4b). Increased PD-L2 expression was also detected in the host compartment, particularly on dendritic cells (Fig. 4c). In addition to increased PD-L2 expression, PD-L1 levels on immune cell subsets were elevated in mice bearing PD-L1 deficient tumour cells. Although small increases in myeloid and lymphoid cells expressing PD-L1 were observed (Fig. 4d), PD-L1 levels on PD-L1 expressing cells were significantly increased on tumour-associated myeloid cells (Fig. 4e; Student's *t*-test $P < 0.05$). Most profound PD-L1 increases in the myeloid compartment were detected in DCs and monocytic MDSCs (Fig. 4e). Granulocytic MDSCs remained among the cell types with highest PD-L1 expression, and their absolute numbers

**Figure 3 | Reconstituting PD-L1 expression in tumour cells influences tumour outgrowth.** PD-L1-deficient MC38 tumour cells were reconstituted with doxycycline inducible RFP (red circles), or PD-L1 (blue circle) using a lentiviral construct encoding constitutive GFP. PD-L1 reconstitution resulted in restored tumour outgrowth (**a**), while PD-L1-deficient tumours with doxycycline inducible RFP regressed. End-point analysis at day 24 showed that mixed cell (red/blue circles) inoculation led to tumours slightly smaller compared to tumours generated from PD-L1-enabled tumour cells only (**b**). While RFP + GFP + tumour cells were detectable by flow cytometry at time of inoculation, this population was lost at endpoint, and solely PD-L1 + GFP + cells remained from the mixed inoculation (**c**). PD-L1 on target cells can inhibit antigen specific cytolytic T-cell activity in a dose dependent manner *in vitro* (**d**). Following complete regression of PD-L1-deficient CT-26 tumours, mice re-challenged with PD-L1-deficient (red) or wildtype (blue) tumours showed complete tumour regression (**e**), despite outgrowth of syngeneic EMT6 (black) tumours on the opposing flank (**f**). Following complete regression of MC38 PD-L1-deficient tumours, mice re-challenged with MC38 tumour cells over-expressing doxycycline induced PD-L1 (light blue) showed near complete tumor regression (**g**), despite over-expressed PD-L1 enabling slightly faster tumour progression in naive mice (**h**). Characterization of tumor subsets by flow cytometry in tumours was performed at day 19 post-inoculation. Data is representative of at least two individual study repeats, with 3 to 10 mice/group. Cytolytic *in vitro* assays were repeated four times with representative data shown. Dox, doxycycline; NA, naive; OVA, ovalbumin. Statistical significance was determined by Student's *t*-test (*$P < 0.05$; **$P < 0.01$; ***$P < 0.001$). Error bars depict s.d. from the mean. Mouse cartoon modified from ref. 15.

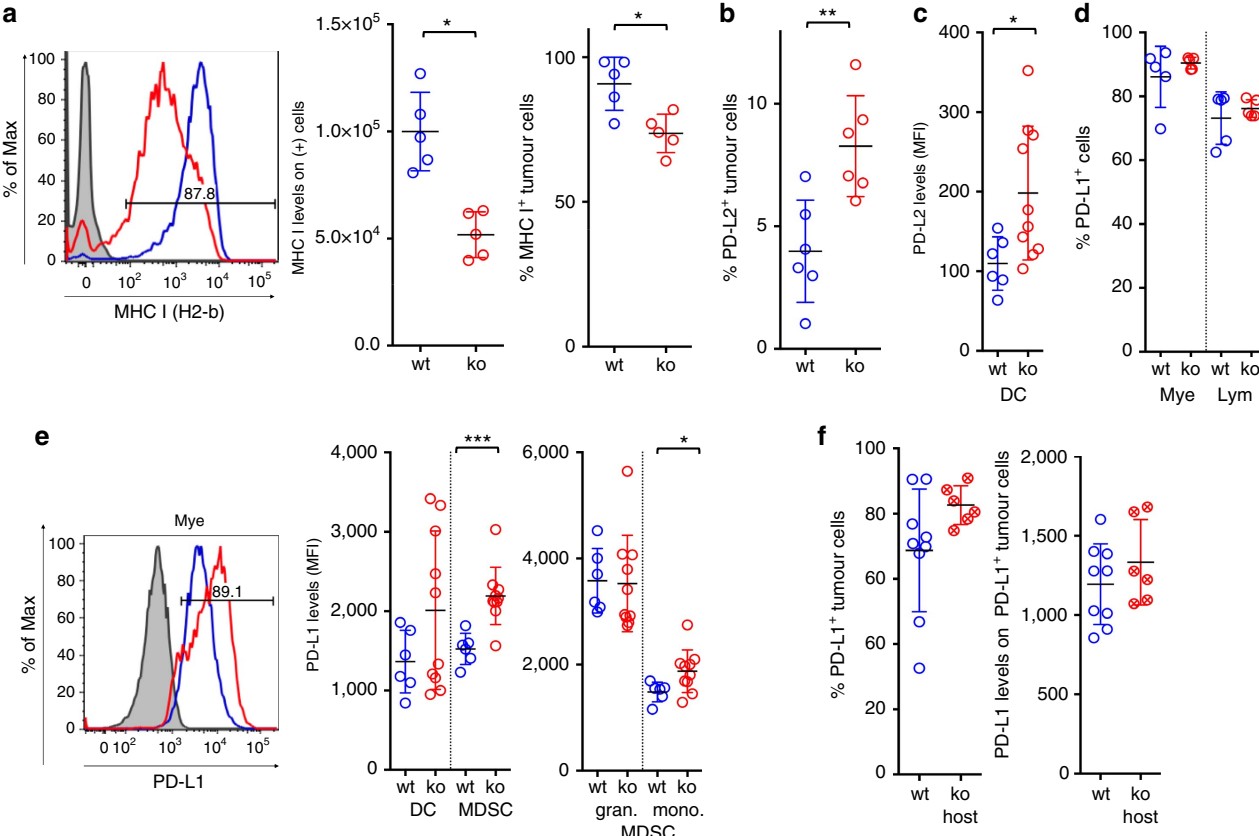

**Figure 4 | Outgrowing tumours lacking PD-L1 apply various putative escape mechanisms.** Flow cytometry analysis of MC-38 tumours indicated that PD-L1-deficient tumours contained fewer MHC-I positive tumour cells and reduced MHC-I surface expression levels (**a**), increased frequency of PD-L2 expressing tumour cells (**b**), and PD-L2 levels on myeloid dendritic cells (DC) (**c**). Although the percentage of PD-L1 positive myeloid and lymphocyte populations were unchanged (**d**), PD-L1 levels were notably increased in monocytic myeloid derived suppressor cells (MDSC) isolated from PD-L1-deficient tumours (**e**). PD-L1 levels on wild-type MC38 tumour cells remained high when implanted in wild-type or PD-L1-deficient hosts (**f**). Data shown is from wt MC-38 tumours in PD-L1-deficient host (crossed circles) or PD-L1-deficient MC-38 tumours in wt host (open circles), with PD-L1 wt status represented in blue and PD-L1 deficiency (ko) represented in red. Characterization of immune subsets by flow cytometry in in vivo tumours was performed at day 24 post-inoculation. Statistical significance was determined by Student's t-test (\*P < 0.05; \*\*P < 0.01; \*\*\*P < 0.001). Error bars depict s.d. from the mean.

were significantly increased (Fig. 2i; Student's t-test P < 0.05). PD-L1 expression on antigen presenting myeloid cells has been implicated in delivering an inhibitory signal during the early priming phase and hence may inhibit T-cell activation[16,21]. It is conceivable that the increased levels of PD-L1 on myeloid cells convey an inhibitory signal to the infiltrating T-cell population. In PD-L1-deficient hosts, increased PD-1 expression on infiltrating T-cells (Fig. 2a) might synergize with the high levels of PD-L1 expression on tumour cells (Fig. 4f) to prevent full regression in a subset of tumours.

Altogether, these data suggest that total levels of PD-L1 expressed on host immune cells serve as a sensitive read out of heightened immune activation within the tumour microenvironment, and that the overall level of PD-L1 from various cellular compartments act to influence the degree of direct anti-tumour immune activity together.

**PD-L1 from tumour and host compartment work in concert.** To test whether increased immune cell PD-L1 expression can be a mechanism of tumour escape in PD-L1-deficient tumours, we compared tumour growth in MC38 models with PD-L1 deficiency on the tumour, the host, and both compartments. In line with previous experiments, PD-L1 loss in the tumour

or host compartment led to tumour regressions (Fig. 5a), with a subset of tumours achieving sustained growth despite the lack of PD-L1 in the host or tumour compartment. However, when neither the tumour nor the host cells expressed PD-L1 to dampen the developing immune response, the highest rate of tumour regressions was observed with near complete prevention of tumour escape (Fig. 5a; meta-analysis of two independent repeat studies in Fig. 5c and Table 1). Antibody blockade of PD-L1 in the context of PD-L1-deficient tumours (Fig. 5b) mimicked the effect seen in the combined tumour/host PD-L1-deficient setting, confirming that PD-L1, more so than PD-L2, was a main driver of immune suppression in the outgrowing PD-L1-deficient tumours. As the frequency of escape was similar between complete genetic deletion of PD-L1 (Fig. 5a bottom right panel) and anti-PD-1 treatment in PD-L1-deficient tumours (Fig. 5b, right panel), escape following complete functional ablation of PD-L1 is most likely driven by mechanisms independent of the PD-1 axis. Highest activation of T-cells as monitored by cytolytic gene expression required the complete deletion of PD-L1 on both host and tumour compartment (Fig. 5d), illustrating the dose-dependent relationship of PD-L1 levels and T-cell activation status. In addition, induction of potential escape mechanisms such as ECM remodelling and EMT observed in PD-L1-deficient

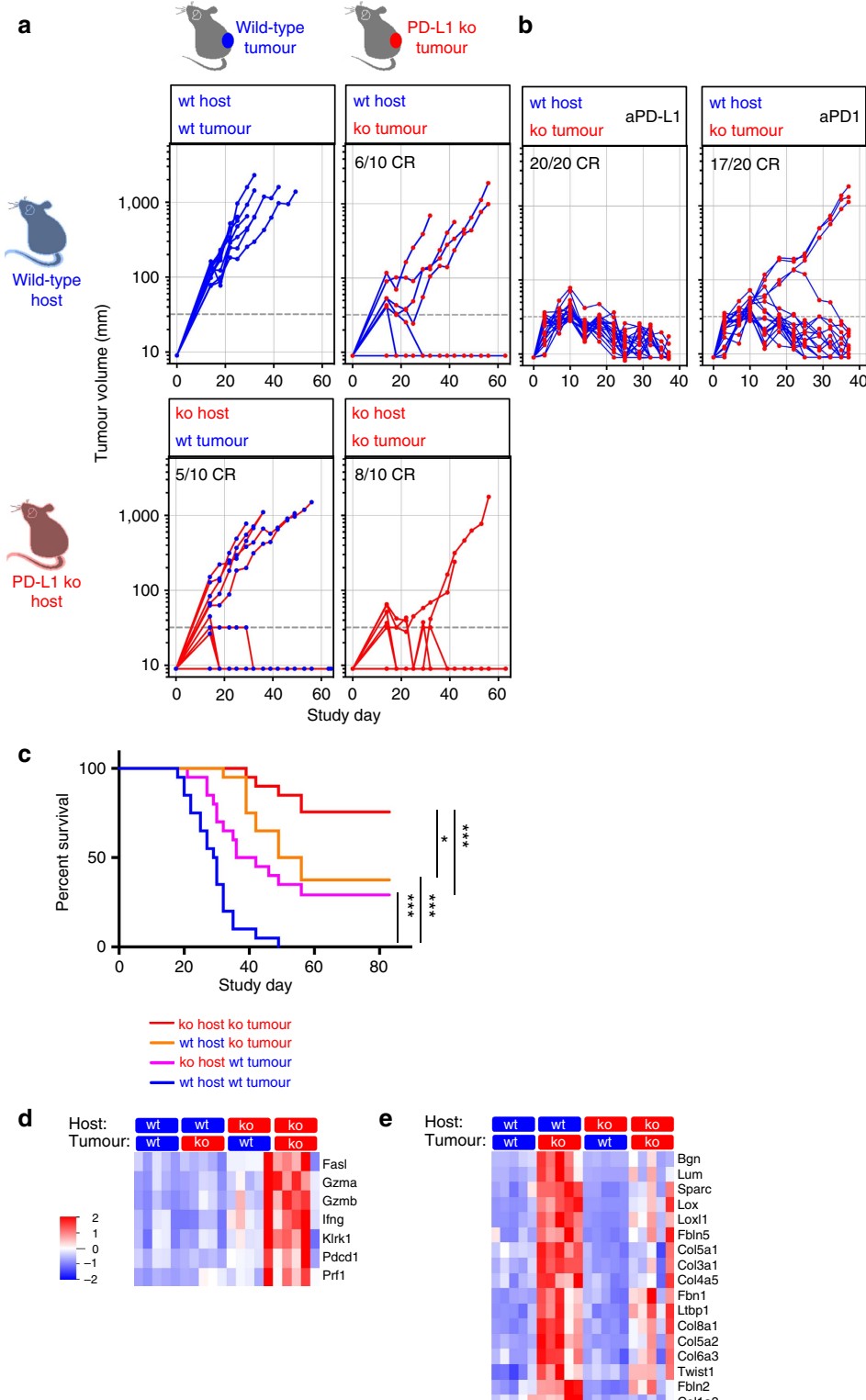

**Figure 5 | Deletion of PD-L1 from both tumor and host compartments lead to most profound frequency of tumor regressions.** Tumour growth and frequency of regressions of wild-type (blue marker) and PD-L1-deficient (red marker) MC38 tumours in wild-type (blue line) and PD-L1-deficient (red line) host mice (**a**), with highest frequency of regressions observed when PD-L1 was genetically ablated in the tumour and the host compartment. Statistical significance was analysed in two repeat studies (**c**: summary of two independent repeat studies with $n = 10$ per group per study; Mantel-Cox log-rank test; detailed statistics shown in table 1). Concomitant treatment of PD-L1-deficient tumours with either anti-PD-L1 or anti-PD-1 blocking antibodies increased frequency of tumour regressions (**b**). Optimal T-cell activation as represented by cytolytic gene expression required deletion of PD-L1 in both tumour and host compartment (**d**), and alleviated degree of potential resistance mechansims represented by stromal remodelling and EMT gene expression (**e**). In tumour growth curves, PD-L1 status of the host is the line colour whereas the tumor status is the symbol colour. Gene expression analysis limited to genes meeting $Q$ value $< 0.05$. Error bars depict s.d. from the mean. Mouse cartoon modified from ref. 15.

**Table 1 | Log-rank (Mantel-Cox) statistical analysis of the two independent repeat studies with PD-L1 WT or KO mice bearing PD-L1 WT or KO MC38 tumours.**

| Genotype | Comparator | Hazards ratio for comparator | P value | 95% CI |
|---|---|---|---|---|
| Host WT Tumour WT | Host WT Tumour KO | 0.19 | <**0.0001** | 0.03 0.17 |
| | Host KO Tumour WT | 0.34 | **0.004** | 0.12–0.49 |
| | Host KO Tumour KO | 0.08 | <**0.0001** | 0.02–0.10 |
| Host KO Tumour KO | Host WT Tumour KO | 3.53 | **0.0141** | 1.37–9.76 |
| | Host KO Tumour WT | 5.31 | **0.0008** | 2.04–13.48 |
| Host WT Tumour KO | Host KO Tumour WT | 1.69 | 0.16 | 0.82–3.93 |

CI, confidence interval; KO, knockout; WT, wild type.
Bold entries depict significant P-values (P<0.05).

tumours (Fig. 2g) were ameliorated with the loss of PD-L1 on both tumour and host compartments (Fig. 5e).

These results support a model in which increased immune cell PD-L1 expression, as a result of heightened immune activation in PD-L1-deficient tumours, is sufficient to allow tumour escape by dampening the cytotoxic activity of T-cells. More importantly, it suggests that PD-L1 from disparate cellular sources, including tumour cells, myeloid or other immune cells can similarly modulate the degree of cytotoxic T-cell function and activity in the tumour microenvironment. Furthermore, PD-L1 expression in both the host and tumour compartment contribute to immune suppression in a non-redundant fashion, and that both could be predictive of sensitivity to therapeutic agents targeting the PD-L1/PD-1 axis.

## Discussion

We have shown that expression of PD-L1 by both tumour and host cells play critical roles in mediating immune suppression of anti-tumour T cell responses. Although elimination of the PD-L1 gene from tumour cells significantly impaired their growth in immune competent hosts, some tumours did escape. In these instances, high PD-L1 expression by infiltrating myeloid cells appeared to provide a compensatory source of the inhibitory ligand. Furthermore, PD-L1-positive tumours implanted in PD-L1-deficient mice also exhibited markedly repressed growth. These findings indicate that infiltrating immune cells play a critical role in negatively regulating T cell responses even at steady state, independent of PD-L1 expression by tumour cells. Although mouse models do not fully recapitulate the complexity of human cancer, they offer a reductionist approach to definitively assess the contribution of tumour- and host-derived PD-L1 in the prevention of anti-tumour immunity. Despite the limitation of mouse models in predicting human disease dynamics, our results offer a potential explanation for the clinical observation that expression of PD-L1 by infiltrating cells is highly correlated with response to anti-PD-L1/PD-1 therapy in human cancer patients. Infiltrating myeloid cells may respond more rapidly and effectively to IFNγ released by T effector cells, thus suppressing initial T cell responses at the level of antigen presentation, including the activation of memory T cells or priming of naive T cells, before the surrounding tumour cells have the chance to react and upregulate PD-L1 themselves. Our results clearly demonstrate that in preclinical models, regulation of T cell function in tumours by PD-L1/PD-1 involves

more than just a simple adaptive response by the tumour cells themselves, but rather involves an intimate and complex interchange between the tumour and its immune microenvironment. These results have potential broad implications for the clinical setting, where measuring PD-L1 expression in both compartments may more faithfully predict therapeutic benefit from antibodies interfering with PD-L1 activity.

## Methods

**Plasmid construction.** Human codon optimized *S. pyogenes* Cas9 was cloned into a pRK vector and expressed via the human cytomegalovirus (CMV) immediate-early promoter. Individual gRNAs (genomic target sites: gRNA-B: 5′-CATAATCAGCTACGGTGGTGCGG-3′; gRNA-D: 5′-AATCAACCAGAG AATTTCCGTGG-3′; gRNA: 5′-GAGTCTGTGTGTTCTCACTTTGG-3′) targeting mouse PD-L1 were cloned downstream of and expressed from the human U6 promoter of the pLKO.5 vector (Sigma, #SHC-201). Lentiviral plasmids were constructed into a pINDUCER11 vector containing constitutive EGFP and a tetracycline response element driving TurboRFP[22]. For inducible PD-L1, mouse PD-L1 cDNA was amplified using primers mPD-L1-F: 5′-AGACTACCGGTC GCCACCATGAGGATATTTGCTGGCATTATATTCACAG-3′ and mPD-L1-R: 5′-CATGTGTCACGCGTTTACGTCTCCTCGAATTGTGTATCATTTCG-3′, and PCR products were cloned using a TOPO 2.1 cloning kit (Life Technologies). Following sequence verification, mouse PD-L1 cDNA was exchanged for the TurboRFP and shRNA cassette in pINDUCER11 via AgeI and MluI (NEB) restriction digest and T4 DNA ligation (NEB). For inducible RFP, the parental pINDUCER11 vector was co-digested with NotI and MluI restriction enzymes (NEB) to remove the shRNA cassette. The linearized vector was subsequently treated with DNA Polymerase I-Klenow (Promega) to blunt end DNA overhangs, followed by T4 DNA ligation.

**Generating PD-L1-deficient cell lines.** Parental (obtained from external vendor such as ATCC, maintained at dedicated internal cell line facility) and PD-L1-deficient MC38 and CT26 (ATCC) colon carcinoma cell lines were maintained in RPMI with 10% fetal bovine serum and 2 mM GlutaMAX (Life Technologies) and routinely tested for mycoplasma contamination (Lonza Mycoalert and Stratagene Mycosensor). Plasmids containing gRNAs and CAS9 were co-transfected into MC38 or CT26 parental cell lines with Lipofectamine LTX with PLUS Reagent (Life Technologies) in Opti-MEM I (Life Technologies). After transient transfection, cells were expanded and stimulated for 48 h with 20 ng ml$^{-1}$ of mouse interferon-γ (R & D Systems) to induce PD-L1 expression before they were single cell sorted on a BD FACS Aria (BD Biosciences). We confirmed PD-L1-deficient clones by flow cytometry and RT-PCR with TaqMan Assays Mm00452054_m1 (PD-L1 01) and Mm03048247_m1 (PD-L1 03; Life Technologies).

**Generating inducible PD-L1 cell lines.** Lentivirus was produced following established protocol[23]. Briefly, low passage HEK-293 cells were transiently transfected with a combination of pINDUCER11 expression plasmid, Δ8.9 packaging plasmid and VSVG envelop plasmid using Lipofectamine 2000 (Life Technologies) in Opti-MEM I media (Life Technologies). After 72 h, lentiviruses were concentrated from cultured supernatants with PEG-it Virus Precipitation Solution (System Biosciences) and titrated. Parental and PD-L1-deficient MC38 and CT26 cell lines were infected at low MOI with inducible PD-L1 lentivirus and polybrene (8 μg ml$^{-1}$). Infected cells were induced with doxycycline (1 μg ml$^{-1}$) for 48 h and then bulk sorted for uniformly induced EGFP+, RFP+ or EGFP+, PD-L1+ populations.

**Syngeneic tumour studies.** *In vivo* tumour studies were performed as follows[24]: age-matched 6–8 week old female Balb/C or C57Bl/6 (Charles River) or Rag2$^{-/-}$ (Taconic) mice were inoculated subcutaneously in the right unilateral flank with $1 \times 10^5$ tumour cells suspended in Hanks's Buffered Saline Solution (HBSS) and phenol red-free Matrigel (BD Bioscience). For studies using doxycycline inducible PD-L1 cell lines, mice and cells were dosed with doxycycline (1 ug/ml) for 48 h before tumour inoculation, and then kept on doxycycline throughout the remainder of study. For anti-PD-L1 treatment studies, mice with tumours starting at ~150 mm3 were treated via intra-peritoneal injection with anti-PD-L1 (clone 6E11), 10 mg kg$^{-1}$, three times per week for one week. For CD8 T-cell depletion studies, mice were treated via intra-peritoneal injection with anti-CD8 (clone 2.43), 25 mg kg$^{-1}$, 1 day before tumour inoculation and then at day 3 and 7 post inoculation. Tumour volumes were measured and calculated twice per week using the modified ellipsoid formula ½ × (length × width$^2$). Tumours <32 mm$^3$ were considered completely regressed, whereas tumours >2,000 mm$^3$ were considered progressed and animals were killed. Similarly, animals whose tumours ulcerated before progression or complete response were killed and removed from study. All animals studies performed were in compliance with protocols approved by the Genentech Institutional Animal Care and Use Committee.

**Re-challenge tumour studies.** After a 2-month tumour-free period, mice with complete regression of PD-L1-deficient tumours, as well as age matched naive control mice, were re-inoculated subcutaneously in the left unilateral flank with either $1 \times 10^5$ wild-type, PD-L1-deficient, or -inducible PD-L1 tumour cells. In certain studies, mice were also inoculated with $1 \times 10^5$ EMT6 breast carcinoma cells in the fourth mammary fat pad on the opposing right side. Tumour volumes were measured and calculated twice per week as described above. Animals whose single tumour volume exceeded 2,000 mm³ or combined tumour volume exceeded 3,000 mm³ were killed. All animals studies performed were in compliance with protocols approved by the Genentech Institutional Animal Care and Use Committee.

**Generation of PD-L1 knockout mice.** We constructed a gene-targeting vector (TNLOX1-3) that by homologous recombination deletes the first two coding exons of the mouse CD274/PD-L1 gene and replace it with a neomycin resistance gene cassette (neo), thereby eliminating the start codon and the IgV domain required for PD-1 binding. Homologous recombination events were screened by PCR and verified by Southern blot analysis. Two targeted embryonic stem (ES) cell lines were injected into C57BL/6 blastocysts. Founder mice showing germ-line transmission were interbred to produce homozygous PD-L1-deficient mice. Mice were screened by PCR to confirm genotype: wild-type (WT), knockout (KO) or heterozygous (HET). Inactivation of the PD-L1 locus was verified by Southern blot analysis and loss of protein expression was confirmed by flow cytometry.

**FACS analysis.** Tissue collection of spleens and draining lymph nodes were in RPMI with 5% fetal bovine serum on ice, and processed through 40 μM cell strainers to make single cell suspensions. Red blood cells were lysed in ACK lysis buffer (150 mM $NH_4Cl$, 10 mM $KHCO_3$, 0.1 mM $Na_2$ EDTA, pH 7.2). Between days 14–24 post-inoculation when tumours reached ∼150–300 mm³, tumours were isolated, homogenized and digested in 1 mg ml$^{-1}$ Collagenase D (Roche) and 0.2 mg ml$^{-1}$ Dnase I (Roche) in RPMI media with 5% fetal bovine serum to generate single cell suspensions for flow cytometry analysis. Cells were stained using standard protocols in Brilliant Stain Buffer (BD Biosciences) for surface markers CD45 (clone 30-F11, BD Biosciences), CD3 (clone 145-2C11, BD Biosciences), CD4 (clone RM4-5, BD Biosciences), CD8 (clone 53-6.7, BD Biosciences), CD25 (clone PC61, BD Biosciences), PD-1 (clone J43, BD Biosciences), CD44 (clone 1M7, BD Biosciences), CD62L (clone MEL-14, BD Biosciences), H-2Kb/H-2Db (clone 28-8-6, Biolegend), H-2Kd/H-2Dd (clone 34-1-2S, Biolegend), PD-L1 (clone MIH5, eBioscience) and LIVE/DEAD Fixable Near-IR Dead Cell Stain (Life Technologies). Cells stained for FoxP3 (clone FJK-165, eBioscience) and/or Ki-67 (clone B56, BD Biosciences) were fixed/permeabilized with the FoxP3/Transcription Factor Staining Buffer Set (eBioscience) before staining for intracellular proteins. Cells were analysed on the BD LSR Fortessa and sorted on the BD FACS Aria. In analysis, dendritic cells (DC) = CD45 +, CD11b +, CD11c +; myeloid derived suppressor cells (MDSC) = CD45 +, CD11b +, CD11c −; granulocytic MDSC = CD45 +, CD11b +, CD11c −, Ly6C +, Ly6G +; monocytic MDSC = CD45 +, CD11b +, CD11c −, Ly6C +, Ly6G −.

**Immunohistochemistry.** Formalin-fixed, paraffin-embedded human tumour samples were obtained through commercial vendor (MT group, Van Nuys, CA, USA) with each patient individually consented in accordance with the Food and Drug Administration good clinical practice guidelines. Samples were processed into 4um tissue sections. Mouse tumours were collected at ∼250 mm³ and similarly processed. Tumour sections were stained with human anti-PD-L1 (clone SP142, Ventana), Ki-67 (ThermoFisher), cleaved caspase 3 or phosphor-STAT3 (Cell Signaling), and counter-stained with hematoxylin on a Ventana Discovery XT instrument. Brightfield whole slide scans were obtained on a Nanozoomer slide scanner (Hamamatsu). Quantification of total number of positive objects per mm² were completed over one section per animal per group, with $n = 5$–7 per group.

**IncuCyte cell growth assay.** Parental and PD-L1-deficient MC38 and CT26 cell lines were plated in triplicate at 1,500 cells per well in a 384 well plate with CellToxTM Green (Promega) at a 1:5,000 dilution. Cells were imaged at ×10 magnification in an IncuCyte Zoom Live-content imaging system (Essen Bioscience) at 37 °C with 5% $CO_2$. Images were acquired every 4 h for one week, with two images per well. Data was analysed using IncuCyte analysis software to detect and quantify live cell confluence (phase-contrast), and dead or dying cells (fluorescence) from the same well. Averages with standard error of the mean at each time point were plotted in Prism (Graphpad).

**Cytolytic assay.** Single cell suspensions of splenocytes collected from C57BL/6-Tg(TcraTcrb)1,100 Mjb/J (OT-1) mice (JAX) were cultured for 5 days with ovalbumin SIINFEKL peptide at 1 ng ml$^{-1}$ with 20 units per ml of mouse IL-2 (Roche) in RPMI with 10% fetal bovine serum, 20 uM HEPES, 55 μM 2-mercaptoethanol, and 1× concentrations of the following supplements from Life Technologies: GlutaMAX, sodium pyruvate, penicillin/streptomycin, and non-essential amino acids. On day 5, primed CD8 + T-cells were purified by negative selection kit (Miltenyi Biotec). MC38 target cells (PD-L1 KO, and inducible PD-L1 (iPD-L1) were pre-treated with or without doxycycline at 1 μg ml$^{-1}$ for 2 days pre-treatment. Surface PD-L1 levels on MC-38 target cells were confirmed by FACS prior to labelling with CFSE at 1 μM (Life Technologies). Labelled MC-38 target cells were co-cultured with purified CD8 + T cells. Co-cultures were set-up in triplicate wells at an E:T ratio of 10:1 in 96 well flat-bottom plates in the presence or absence of SIINFEKL at 0.001 or 0.0001 ng ml$^{-1}$ overnight before flow cytometric analysis of CFSE plus propidium iodide (PI) (BD Biosciences). The percent lysis was calculated as follows: % lysis = [%CFSE + PI +]/[total CFSE +] × 100.

**RNA expression analysis.** Total RNA was isolated from fresh cultured cells or frozen tumour samples at d9 post implantation (Figs 2 and 4) or median tumour size of 250 mm³ (Supplementary Fig. 6) using RNeasy (Qiagen), including an on-column DNase I digestion. cDNA was prepared using High Capacity cDNA Reverse Transcriptase Kit (Life Technologies). Quantitative RT-PCR was performed with the ABI 7900 HT Fast Real-Time PCR System (Applied Biosystems). Gene expression data is normalized to three control genes (ACTB, RPS13 and HMBS; see Supplementary Table 1).

**RNA-Seq analysis.** Total RNA was extracted from MC38 derived in vivo tumours at median tumour volume of 250 mm³ using RNeasy (Qiagen) as described above. RNA integrity and concentration of RNA samples were determined respectively by Agilent 2100 Bioanalyzer (Agilent Genomics), Fragment Analyser (Advanced Analytical Techonologies) and NanoDrop 8000 (Thermo Scientific) before their processing by RNA-seq. 1 μg of total RNA was used as input material for library preparation using TruSeq RNA Sample Preparation Kit v2 (Illumina). Size of the libraries was confirmed using Fragment Analyser (Advanced Analytical Technologies) and their concentration was determined by qPCR-based method using Library Quantification Kit (KAPA). The libraries were multiplexed and then sequenced on Illumina HiSeq2500 (Illumina) to generate ∼25 million uniquely mapping reads per sample. Reads were mapped to mm⁹ using GSNAP[25].

**Differential gene expression.** Differential expression analysis between knockout and wild-type lines was conducted using the voom/limma pipeline[26,27]. Briefly, lowly expressed genes were first removed by only keeping genes which were expressed at ≥ two counts per million (CPM) in > = six samples. The filtered count matrix was then sent through voom to model the mean/variance trend of the read counts before differential expression analysis. All statistics reported from the RNASeq data in this manuscript are the FDR corrected P values (Q values).

**Gene set enrichment analysis.** Gene set enrichment analysis (GSEA) between PD-L1 knockout and WT tumours was performed using CAMERA[28,29] against the MSigDB (v5.0; ref. 30) and Panther[31] GOSLIM gene sets using a pre-specified inter-gene correlation value set to 0.02. These results are included in Supplementary Data 1 (hWT_tKO versus hWT_tWT comparison)and 2 (hKO_tWT versus hWT_tWT comparison). The gene sets chosen for display in Fig. 2c,g were selected by (a) summarizing the log fold change of each gene set by the average log fold change of its constituent genes; (b) sorting the gene sets by decreasing average log fold change; and finally (c) filtering them down to include only the top 15 MSigDB hallmark and REACTOME (MSigDB c2) gene sets with a CAMERA FDR < 0.05. To visualize the activity of a gene set per sample in the heatmap, we generated single sample gene set scores as described[32] and plot their row-wise z-transformed values.

**PD-L1 and PD-1 binding assay.** Mouse PD-1-Fc was biotinylated with EZ-Link sulfo-NHS-LC-biotin (Pierce) for 30 min at room temperature as described by the manufacturer. Excess non-reacted biotin was removed with Quick Spin High Capacity Columns, G50-Sephadex (Roche) as described by the manufacturer. Nunc Maxisorp 384 well plate was coated with 250 ng ml$^{-1}$ mPD-L1-Fc in PBS overnight, washed with 0.05% Tween-20 in PBS and blocked with 0.5% BSA. Wild-type and PD-L1-deficient MC38 or CT26 clones were treated with or without IFNγ (20 ng ml$^{-1}$, R&D Systems) for 48 h. Supernatants were collected, serially diluted and incubated to test for binding to biotinylated mouse PD-1-Fc (250 ng ml$^{-1}$). Plates were washed and visualized by Streptavidin-HRP (1:20,000, GE Healthcare) and TMB (Kirkegaard and Perry Laboratories) at 450 nm absorbance. Anti-mouse PD-L1 (clone 6E11) and recombinant mouse PD-L1-Fc protein were used (starting at 15 μg ml$^{-1}$) as a positive control for binding to mouse PD-1.

**Data availability.** The RNASeq data have been de[posited at the NCBI Sequence Read Archive (SRA) under BioProject ID PRJNA356678. All other data supporting the findings of this study are available within the article and its Supplementary Information files and from the corresponding author on reasonable request.

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

## Acknowledgements

We thank Merone Roose-Girma, Zora Modrusan, the IHC and FACS labs, and the vivarium team for expert technical assistance. We thank Shannon Turley, Jill Schartner, Lelia Delamarre and Jing Qing for valuable discussions.

## Author contributions

J.L. performed and analysed genetic ablation experiments and immune profiling of *in vivo* samples; L.S. performed RNA isolations and analysis; J.C. performed *in vitro* cytotoxicity assay; J.C. and R.H. performed the wt/ko host/tumour compound experiments; A.N. and V.J. performed all other *in vivo* experiments, K.T. supervised and coordinated all *in vivo* experiments; B.H. designed the CRISPR and transgene constructs; S.L. was responsible for all bioinformatics analysis including RNASeq; P. C. supervised histological analysis of tumour tissue; J.M. oversaw generation of the PD-L1 host ko mice; J.E.-A. performed image quantification analysis on histological sections; R.Y. and B.I. provided input on initial project conception; M.B. and I.M. provided input and institutional support of the project; M.S. wrote the manuscript; J.G., M.B., I.M., J.L., J.K. edited the manuscript; J.K. designed and oversaw the *in vitro* cytotoxicity as well as *in vivo* experiments involving PD-L1 host ko mice; M.S. designed, interpreted and oversaw this study.

## Additional information

**Competing financial interests:** All authors are current or former employees of Genentech Inc., a member of the Roche group.

