## [Peer Review File · Nature Communications]

REVIEWERS' COMMENTS:

Reviewer #2 (Remarks to the Author):

The authors have made changes that improve the article. Here are minor additional things to consider:

- In page 2 there is reference to a submitted article by Kowanetz et al and also to an ESMO 2016 presentation, but these citations do not seem to comply with the reference guidelines from the journal.

- It should be clear in the conclusions of this article that they refer to the author's mouse system. The role of PD-L1 in different cell types described in this work should not be extrapolated to humans on treatment with anti-PD-1/L1 therapies as this is not being tested here. As the mouse systems are different than what is being noted in humans (for example, response to anti-PD-1/L1 in MC38 depends on both CD4 and CD8 T cells, which may be different from what is evident from human tumor biopsies responding to this mode of therapy), then the authors should be careful in how they state their conclusions.

RESPONSE to REVIEWERS COMMENTS

We would like to thank reviewer #2 for reading our updated manuscript and again providing valuable insight how to improve the manuscript. We have addresses the remaining concerns and suggestions as listed below:

Reviewer #2 (Remarks to the Author):

The authors have made changes that improve the article. Here are minor additional things to consider:

- In page 2 there is reference to a submitted article by Kowanetz et al and also to an ESMO 2016 presentation, but these citations do not seem to comply with the reference guidelines from the journal.

We have removed the reference to the ESMO presentations and the manuscript by Kowanetz et al., as that manuscript still remains under review at Nature and has not been published yet.

- It should be clear in the conclusions of this article that they refer to the author's mouse system. The role of PD-L1 in different cell types described in this work should not be extrapolated to humans on treatment with anti-PD-1/L1 therapies as this is not being tested here. As the mouse systems are

different than what is being noted in humans (for example, response to anti-PD-1/L1 in MC38 depends on both CD4 and CD8 T cells, which may be different from what is evident from human tumor biopsies responding to this mode of therapy), then the authors should be careful in how they state their conclusions.

Our work is focusing on the reverse translation of an correlative observation made in the clinic, namely that PD-L1 expression on tumour and immune cells can be independent, correlative predictors of response to PD-L1/PD-1 blocking antibodies. We opted for the mouse system as these preclinical models are amenable to genetic manipulation and hence allow mechanistic evaluation of the question based on these clinical correlates. However, we share the reviewer's caution on how to use preclinical data in predicting drug efficacy and extrapolating biology to the human setting. We have added language to further stress that our data was generated in mouse model systems.